# Blind Analysis of Food-Related IgG Identifies Five Possible Nutritional Clusters for the Italian Population: Future Implications for Pregnancy and Lactation

**DOI:** 10.3390/nu11051096

**Published:** 2019-05-17

**Authors:** Gabriele Piuri, Enrico Ferrazzi, Attilio Francesco Speciani

**Affiliations:** 1Inflammation Society, 18 Woodlands Park, Bexley DA52EL, UK; gabriele.piuri@me.com (G.P.); enrico.ferrazzi@unimi.it (E.F.); 2Fondazione IRCCS Cà Granda, Ospedale Maggiore Policlinico, Department of Clinical Sciences and Community Health, University of Milan, 20122 Milan, Italy

**Keywords:** pregnancy, lactation, weaning, food-related IgG, food clusters, non-IgE-mediated food reactions

## Abstract

Background: The influence of diet in pregnant women on the immune tolerance process is intricate. Food-specific immunoglobulin G (IgG) was associated with exposure to particular food antigens. The IgG antibodies can cross the placental barrier and enter into the colostrum, and maternal IgG is amply present in breast milk. This justifies studying the immunological connection between food-specific IgG antibodies and the mother–fetus relationship. This study was designed to analyze food-specific IgG concentrations and possible food-specific IgG concentration clusters in a large cohort of subjects with a common food culture. Methods: Food-specific IgG antibody concentrations were detected in 18,012 Caucasian or Southern European subjects over 18 years of age. We used an unsupervised hierarchical clustering algorithm to explore varying degrees of similarity among food-specific IgG antibodies. Results: We identified five food groups by the evaluation of food-specific IgG values: one includes foods with a high nickel content, the second cluster is associated with gluten, the third cluster includes dairy products, the fourth one is connected to fermented foods, and the last group is correlated with cooked oils. Discussion: The knowledge derived from studying a large sample allows us to determine food-specific IgG values from a single pregnant woman, compare it to an epidemic standard, and establish modifications required in her lifestyle to modulate her nutritional habits.

## 1. Introduction

Recently published reviews support the biological association of food-specific immunoglobulin G (IgG) with exposure to particular food antigens [1,2]. IgG antibodies, especially IgG4, are highly related to the immune tolerance of food antigens, and IgG can be responsible for the attenuation of the immunological response. These protective antibodies might regulate the immune response by avoiding the binding of the antigen to the IgE already present on the surface, or by inhibiting the maturation of the dendritic cells, preventing the activation of possible allergic pathways [3]. IgG4 antibodies have diverse anti-inflammatory actions compared to other IgG subclasses. One of these differences is the reduced capacity to induce complement activation due to a low affinity for the C1q and Fc receptors. Due to their ability to bind different antigens in diverse sites, these antibodies do not precipitate antigens [4].

The relationship between IgE-mediated anaphylaxis and more subtle hyper-immune responses associated with high IgG titer is being increasingly studied and has now been precisely defined. Antigens can cause systemic anaphylaxis through the classical pathway by cross-linking IgE bound to mast cell FcεRI or through the alternative path by forming complexes with IgG. New studies have shown an IgG-dependent mechanism of anaphylaxis that involves IgG, FcγRs on macrophages, basophils and neutrophils, complement-derived peptides C3a and C5a, B-cell activating factor (BAFF), and platelet-activating factor (PAF) [5]. According to Finkelman [6,7,8] and Muñoz-Cano et al. [9], IgG, macrophages, basophils, and neutrophils may even be involved in human anaphylaxis. This delicate blinding of food antigens can be dysregulated by an excess of these specific food antigens related to IgG and activate inflammatory processes [8].

This fine-tuned immune equilibrium permits alien food antigens to land safely on the gut mucosa as they are processed and allowed to pass through the mucosal surface and become nutritional elements for the microbiota, becoming accepted as “legal aliens”. This complex recognition occurs on the most notable large external surface in the human body, which is 200 times wider than skin. The repetitive and continuous consumption of a particular food that corresponds to a higher production of specific antibodies can fracture the equilibrium, increase inflammatory mediators [8], and damage the delicate gap junctions in the gut mucosa [10].

High concentrations of food-specific IgG antibodies have been successfully identified as foes in non-IgE-mediated food reactions. Bentz et al. reported significantly higher food-specific IgG antibodies in patients with Crohn’s disease in contrast with healthy controls [11]. The same patients showed significant clinical improvement in inflammatory bowel disease (IBD) symptoms when the foods associated with highly specific IgG were removed from the diet [11]. In agreement with these findings, Cai et al. observed high levels of IgG antibodies to specific food antigens in patients affected by IBD [12]. Alpay et al. evaluated the effect of a personalized nutritional approach based on food-specific IgG in sequences of migraine attacks in a randomized, double-blind, cross-over, headache-diary-based trial on patients diagnosed with migraines without auras [13]. A similar double-blind, placebo-controlled, dietary re-challenge trial was performed by Biesiekierski [14] to prove that the symptoms of irritable bowel syndrome (IBS) could be reduced in patients without proven celiac disease and with clinical sensitivity to gluten. A similar study demonstrated that clinical symptoms of IBS were significantly reduced in patients affected by Sjogren’s syndrome who underwent a diet characterized by high concentrations of food-specific IgG [15]. IBS is a less-defined pathological condition, yet increased gastrointestinal permeability and gut inflammation were observed by Shulman in children [16]. Similar patterns of food-specific IgG and IgG4 were observed in eosinophilic esophagitis [17], and such deposits of IgG4 may even distinguish patients with gastro-esophageal reflux disease versus eosinophilic esophagitis.

The Western diet is prone to causing this imbalance [18] and is characterized by a lack of variety and seasonality that could elicit excessive antibody production against food antigen clusters. In parallel, the lack of fiber in a diet produces a poor prebiotic environment, directly affecting the gut microbiota. These conditions might convert the role of the microbiota from an anti-inflammatory partner of dendritic cells in the intestinal mucosa into an aggressive collection of pro-inflammatory bacteria.

We hypothesized that high food-specific IgG concentrations in subjects suffering from organ or systemic inflammation might be critical in a pro-inflammatory scenario. This study was designed to analyze food-specific IgG concentrations and possible food-specific IgG concentration clusters in a large cohort of subjects with a common food culture and affected by gastroenteric inflammatory symptoms and/or other tissue or systemic inflammations.

## 2. Materials and Methods

### 2.1. Subjects

All of 18,012 patients suspected a relationship between food intake and symptoms such as bloating, constipation, diarrhea, heartburn, gastritis, or other signs and symptoms of possible gastroenteric inflammation. They were all Caucasian or Southern European subjects over 18 years of age. Written consent to this observational study was obtained by all participants who self-enrolled in the study, according to privacy regulations. Sample collection was performed by finger pricks using a nylon swab for the storage of dried blood (Copan Diagnostics, Inc., Murrieta, CA, USA). Blood samples were delivered to a laboratory (Diagnostica Spire S.r.l. Reggio Emilia, Italy) and analyzed within less than seven days from sample collection to ensure the data accuracy. Food-specific IgG antibody concentrations were detected by employing ELISA plates produced by Immunolab GmbH (Kassel, Germany) customized for 44 common Italian food antigens. Lower and upper detection limits for IgG antibodies were 0.0 U/mL and 100 U/mL, respectively.

### 2.2. Statistical Methods

The frequency distribution of food-specific IgG concentrations in this population was determined for each food antigen tested. Non-parametric descriptive statistics (median and interquartile range) were adopted for non-normal (D’Agostino and Pearson normality test) or bimodal distributions.

We used an unsupervised hierarchical clustering algorithm to explore varying degrees of similarity among food-specific IgG, adapting the statistical methods used by Eisen et al. [19]. The object of using this algorithm was to compute a dendrogram that assembled all food-specific IgG into a single tree. For any food-specific IgG, an upper-diagonal similarity matrix containing similarity scores for all pairs of food-specific IgG was calculated. The matrix was scanned to identify the highest value, representing the most similar pair of food-specific IgG. A node was created joining these two foods, and a new food-specific IgG level was computed for the node by averaging observations for the joined elements. The similarity matrix was then updated with this new node, replacing the two joined elements, and the process was repeated (n − 1) times until only a single element remained. The algorithm initially had 44 food-specific IgG antibodies (one for each antigen), which were gradually grouped together into higher-degree clusters according to their similarities. The algorithm stopped when all food-specific IgG antibodies were grouped into the same cluster. The bar on the left side of the dendrogram indicates the dissimilarity (1 − Correlation) for every node. A lower dissimilarity suggests a higher correlation between the foods, or the clusters of foods, linked together by a node and can be useful to better understand the relationship between the different foods.

## 3. Results

Food-specific IgG values were measured in 18,012 Italian patients (74.1% women, mean age 44.5 ± 15.5 years). Table 1 shows the IgG concentration for each of the 44 food antigens tested in alphabetical order and the type of distribution observed. Most IgG distributions were asymmetrical. For food-specific IgG with a bimodal distribution, we indicate medians with an interquartile range of both modes. Due to exposure to different food antigens, the IgG concentrations were very low for foods that showed levels of IgG <1 U/mL, such as olives, peaches, tea, honey, red grapes, and zucchini. High levels were found for foods that showed concentrations of IgG >20 U/mL, such as processed cheese, cow milk, and common wheat. Notably, the IgG values with a bimodal distribution were higher compared with IgG values with an asymmetrical distribution.

Figure 1 shows four models with both asymmetrical and bimodal frequency distribution. We provide diagrams of food-specific IgG values of all 44 foods tested in the Appendix A. Figure 2 shows the dendrogram produced from the results of the clustering algorithm, which identifies five food groups by the evaluation of food-specific IgG values (for every node *p* < 0.001). The first group includes foods with a high nickel content, such as tomato, kiwi fruit, peanuts, almonds, and buckwheat. Inside this group, a second cluster can be identified that includes wheat and associated grains such as Kamut, spelt, and barley. The third cluster includes dairy products (such as cow’s and goat’s milk as well as Parmesan, mozzarella, and ricotta cheese). The fourth one includes yeasts such as *Candida albicans* and *Saccharomyces cerevisiae* and porcini and champignon mushrooms. This cluster is likely connected to fermented foods. The last group contains roasted nuts (peanuts and almonds) and is probably correlated with heated and cooked oils.

## 4. Discussion

### 4.1. Main Findings and Interpretation

The dendrogram obtained by the blind post-hoc matrix, correlated by similarity, allowed us to observe significant correlations between food-specific IgG antibodies that seem to correspond to typical Italian food habits [20,21,22]. Goat’s and cow’s milk, processed cheese, and mozzarella and Parmesan cheese were strongly correlated. The first node between processed cheese and ricotta cheese had the highest similarity. The second highest similarities were found via the dendrogram algorithm and were between mozzarella and Parmesan cheese and between goat’s and cow’s milk. These strict statistical similarities between IgG concentrations were found within dairy products. Such strict correlations, apparently biologically based, were also observed in yeasts (yeast and mushrooms) and in cereals containing gluten. The latter group is probably part of a larger cluster of foods containing nickel. The dendrogram algorithm also segregated a larger group of less strict correlations ranging at a dissimilarity level of 0.8 instead of the 0.3–0.5 dissimilarity level of the previously described clusters. A possible common denominator of these food-specific IgG could be represented by roasted nuts (peanuts and almonds), which are correlated with cooked oils.

The second relevant finding of this large cohort is represented by the distribution of the 44 food-specific IgG levels measured. As expected, the vast majority showed a modal distribution with a marked skewness toward lower concentrations. In eight of the food-specific IgG antibodies, the distribution was bimodal. These latter IgG levels showed significantly higher concentrations compared with IgG concentrations with mono-modal asymmetrical distribution. IgG levels for milk in this Italian population were among these bimodal distributions. The first Gaussian likely represents occasional consumers of milk and dairy products, whereas the second Gaussian reflects daily consumers. Although the use of dairy products is widespread in Italy, a large part of the adult Italian population does not consume milk regularly due to actual or suspected lactase non-persistence [23]. In contrast, the low concentration of specific IgG antibodies for peanuts reflects the nutritional habits of Italians, the vast majority of whom eat peanuts only occasionally.

The differences between IgG levels for different foods could be explained either by eating behaviors connected with increased or reduced intake or by the immunogenicity of the specific food antigen [1,2]. According to these tenets, the difference in the absolute values of IgG for consumers of large amounts of wheat (median concentration 18.22 U/mL) and for consumers of large amounts of honey (median concentration 0.07 U/mL) could be explained by intrinsic antigenic properties.

The production of food-specific IgG antibodies is directly related to the recurrent or prevalent intake of specific foods and can cause an immune reaction inducing, under specific conditions, an increase in inflammatory mediators [8]. The immune system does not specifically recognize foods by IgG, as in the case of IgE, but rather with an approach of similarity, identifying food antigen clusters that reflect the eating habits within different populations. According to this knowledge, the evaluation of the different distribution curves of IgG levels in the Italian population allows us to better understand the possible role of IgG production and how these antibodies can highlight a specific nutritional excess of a food cluster, suggesting a different dietary approach to inflammatory diseases. In agreement with Finkelman [6,7,8], the increased amount of contact with antigenic foods leads to the production of food-specific IgG for a large number of different foods, thus avoiding an absolute prevalence of a single specific antibody to a specific food group. With this information, the activation of a reasonable inflammatory response can be possibly prevented or modulated.

### 4.2. Speculations and Further Hypothesis for Future Implications in Pregnancy and Lactation

The influence of diet in pregnant women on the immune tolerance process is intricate. Thus, diet must be analyzed, not only from a compositional standpoint but also from an evolutionary point of view. Pregnancy symptoms, such as nausea and vomiting in the first trimester, might protect both the pregnant woman and the embryo from potentially harmful substances present in food [24], while, in the last trimester, cravings for food expose the fetus to a higher number of antigens and increase immune contact with the outside environment, yielding an immune imprinting. The fetus’ immune knowledge of the outer world is guaranteed by the IgG produced by the mother. The IgG antibodies can cross the placental barrier and enter into the colostrum, and maternal IgG is amply present in breast milk. This justifies studying the immunological connection between food-specific IgG antibodies and the mother–fetus relationship [5]. This intrauterine information has allowed mammals to regard specific foods as sources of energy. The body knows if the food is sufficient for intimate contact with the organism and uses it as a nutrient.

### 4.3. Strength and Limitations

To the best of our knowledge, this is the first study of food-specific IgG involving thousands of subjects who have the same macro-ethnicity—Caucasians of Southern European ancestry—and who self-enrolled based on common gastro-enteric symptoms. Data on over 18,000 subjects were blindly analyzed by an independent third party that was unaware of the meaning of the codes attributed to each food-specific IgG. A major limitation of the statistical evidence observed is represented by the heterogeneity of the population recruited and by the fact that a direct correlation with different signs and symptoms was not available post hoc. However, the coherence of the distribution of IgG concentrations and the coherence of the clusters identified by the dendrogram algorithm with respect to the Italian food consumption profile allowed us to consider our findings genuine in the ongoing debate on food antigens and gastro-enteric IgG production, immune tolerance, and inflammation.

## 5. Conclusions

The dendrogram algorithm appeared to operate in a process resembling the immune system recognition of food antigens and similar related antigens. This finding allowed us to observe, confirm, and better specify the existence of five large clusters of different foods that appear to react in the same way to IgG antibodies.

## Figures and Tables

**Figure 1 nutrients-11-01096-f001:**
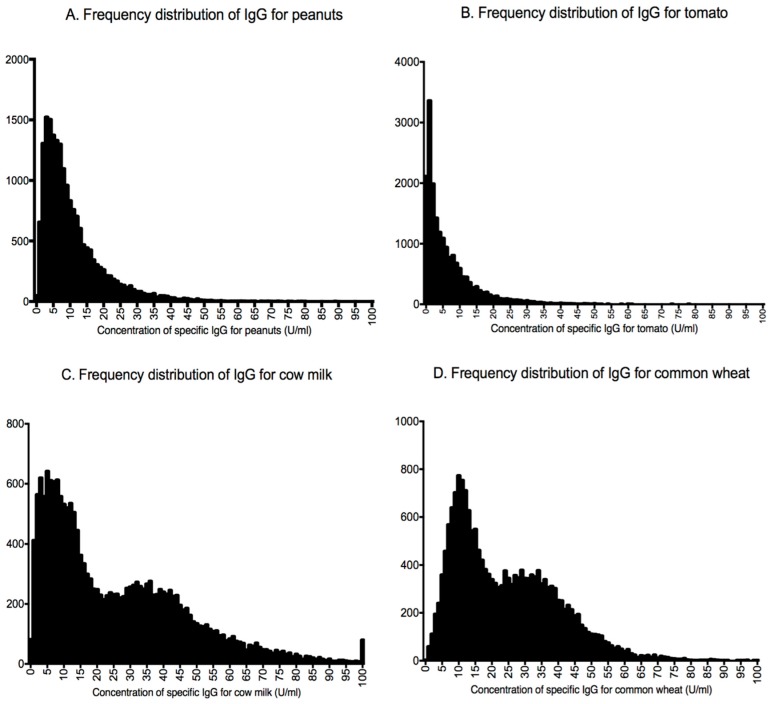
Examples of asymmetrical and bimodal distribution of frequencies for food-specific IgG. (**A**,**B**) Graphs showing two examples (for peanuts and tomato, respectively) of asymmetrical distribution of IgG. (**C**,**D**) Graphs showing two examples (for cow’s milk and common wheat, respectively) of bimodal distribution of IgG.

**Figure 2 nutrients-11-01096-f002:**
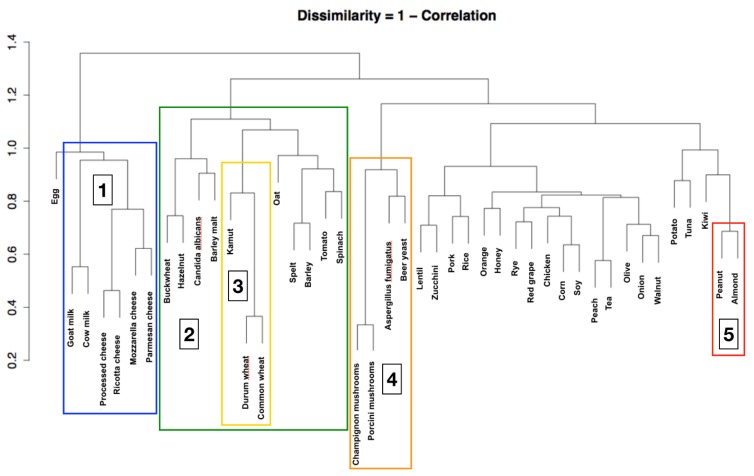
Dendrogram resulting from the clustering algorithm. It is possible to identify five large food clusters: (1) milk and dairy products, (2) foods with high content of nickel, (3) wheat-related grains, (4) fermented foods, and (5) roasted nuts and cooked oils (for every node *p* < 0.001). The bar on the left side of the dendrogram indicates the dissimilarity (1 – Correlation) for every node.

**Table 1 nutrients-11-01096-t001:** List of tested foods, median and interquartile range of corresponding immunoglobulin G (IgG) values, and type of frequency distribution. For food-related IgG with a bimodal distribution, the interquartile range of both modes is indicated.

Food–Antigen	IgG Levels U/mL	Distribution
Almonds	8.35 (4.55–14.25)	Asymmetrical
*Aspergillus fumigatus*	3.95 (1.63–10.75)	Asymmetrical
Barley	9.16 (5.06–15.36)	Asymmetrical
Barley malt	1.17 (0.55–2.54)	Asymmetrical
Buckwheat	6.99 (3.43–12.49)	Asymmetrical
*Candida albicans*	2.48 (0.97–7.00)	Asymmetrical
Canned tuna	3.24 (1.42–6.21)	Asymmetrical
Champignon mushrooms	4.05 (1.38–9.60)	Asymmetrical
Chicken	0.78 (0.29–1.60)	Asymmetrical
Corn	1.32 (0.67–2.51)	Asymmetrical
Eggs	8.63 (3.01–21.58)	Asymmetrical
Goat milk	8.18 (3.92–17.39)	Asymmetrical
Hazelnuts	5.19 (2.46–9.5)	Asymmetrical
Honey	0.07 (0.00–0.41)	Asymmetrical
Kamut	6.47 (3.87–11.27)	Asymmetrical
Kiwis	3.16 (1.55–6.72)	Asymmetrical
Lentils	0.65 (0.25–1.51)	Asymmetrical
Mozzarella cheese	8.71 (4.7–15.56)	Asymmetrical
Oats	1.97 (0.99–4.28)	Asymmetrical
Olives	0.01 (0.00–0.16)	Asymmetrical
Onion	0.71 (0.32–1.40)	Asymmetrical
Oranges	0.19 (0.02–0.79)	Asymmetrical
Parmesan cheese	8.77 (3.45–18.72)	Asymmetrical
Peaches	0.01 (0.00–0.16)	Asymmetrical
Peanuts	7.83 (4.28–13.71)	Asymmetrical
Porcini mushrooms	2.83 (1.1–7.08)	Asymmetrical
Potato	3.34 (1.56–6.30)	Asymmetrical
Red grapes	0.08 (0.00–0.50)	Asymmetrical
Rice	0.52 (0.10–1.48)	Asymmetrical
Ricotta cheese	13.71 (7.32–28.15)	Asymmetrical
Rye	1.28 (0.61–2.65)	Asymmetrical
Soy	0.97 (0.45–2.05)	Asymmetrical
Spinach	0.20 (0.01–0.97)	Asymmetrical
Tea	0.02 (0.00–0.19)	Asymmetrical
Tomato	3.94 (1.20–9.10)	Asymmetrical
Walnuts	0.41 (0.07–1.19)	Asymmetrical
Zucchini	0.08 (0.00–0.60)	Asymmetrical
Beer yeast	3.80 (1.87–6.81) 17.92 (9.01–30.36)	Bimodal
Common wheat	11.07 (7.95–14.61) 32.53 (24.83–41.21)	Bimodal
Cow milk	8.54 (4.68–13.10) 37.55 (27.60–49.70)	Bimodal
Durum wheat	12.04 (7.86–16.47) 34.64 (28.39–42.74)	Bimodal
Pork	1.17 (0.53–2.75) 10.74 (6.05–18.16)	Bimodal
Processed cheese	14.72 (8.86–21.90) 47.19 (38.76–58.96)	Bimodal
Spelt	6.42 (3.71–9.89) 23.46 (18.01–30.26)	Bimodal

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
