# Peer review of "Blind Analysis of Food-Related IgG Identifies Five Possible Nutritional Clusters for the Italian Population: Future Implications for Pregnancy and Lactation"

_nutrients, 2019, doi:10.3390/nu11051096_

Reviewer 1 Report

The present study is very well written and data are well presented. However, the significance of the data and relevance with gastrointestinal disoredrs is rather week. 

Major limitations

1.      The aim of the study which was potential associations of food specific IgGs and gastroenteric 90 inflammatory symptoms and/or other tissue or systemic inflammations, was not shown. Only descriptive data of subjects with rather vague gastroenteric symptoms are described.

2.      Although the abstract starts with significant consideration in pregnant’ women diet, the study included general population and further referral of potential relevance with pregnant women is only as assumption in the discussion.  

Introduction

Lines 76-86: not directly related to the article. Minimize or delete

Subjects:

Line 94: rephrase so that it makes sense

A certain amount of the discussion section refers to speculations and further hypothesis which is not directly supported by the findings: could be minimized

Author Response

We thank you for your revision. Your suggestions are constructive and can improve the quality of our work. We follow the order of your comments to answer to you.

1) The subjects enrolled in this study decided to participate because of their self-perception of a relationship between food intake and symptoms such as bloating, constipation, diarrhea, heartburn, gastritis, or other signs and symptoms of possible gastroenteric inflammation. Gastroenteric symptoms are the most common manifestations that patients connect to food intake. The repetitive and continuous consumption of a particular food determines a higher production of food-specific antibodies, and they can break the tolerance equilibrium, increase inflammatory mediators, and damage the delicate gap junctions in the gut mucosa.

2) Pregnancy does not affect the production of food-specific IgG and the diet in pregnant women is not so different from the diet of the general population. Moreover, the concentration of food-specific IgG remains constant for weeks before any change in nutritional habit affect it. For all these reasons, the general population can well describe the food-specific IgG concentration also evaluable in pregnant women, and the food clusters obtained in the general population are valid even in pregnancy.

As suggested in our study, the increased variety of food intake in the maternal diet could contribute positively to the tolerance of the fetus. In clinical practice, however, for a long time it has been incorrectly recommended to pregnant women to eliminate from their diet the most allergenic foods, favoring a further reduction of food variety. A repetitive diet is likely to lead to a higher production of IgG antibodies to food in response to the significant intake of foods belonging to the same great food clusters. The subsequent continuous consumption of these foods is likely to activate the IgG-mediated pathway with the enhancement of inflammation and, consequently, the reduction of immune tolerance of both mother and baby.

At the best of our knowledge, this is the first time in which the benefit of a varied diet also in pregnancy is, considering food-specific IgG as a food tolerance index. According to this, we decided to insert in the discussion the paragraph "speculations and further hypothesis".

3) Introduction: we have minimized this paragraph as you suggested.

4) Subjects: we have rephrased the sentence.

5) We hope to have already responded to you at the second point.

Linguistic note: before the submission to Nutrients and to you, we have proposed our paper to MDPI (the linguistic service of Nutrients) for revision and for plagiarism, accepting all their notes and comments and fixing them in accordance with their suggestions. 

Thank you for your work. We remain at your disposal for any other suggestions.

Reviewer 2 Report

The article by Piuri and coworkers describes the analysis of food-specific IgGs in samples from over 18,000 subjects and identification of different food groups. The authors suggest an important role of diet in pregnancy and its effects in developing the fetus' immune system. While it is difficult to derive a direct correlation of diet with fetal outcome, given the ad-hoc nature of this analysis from dried blood spots. Nevertheless, the manuscript highlights some interesting findings that could be used as a basis for further analyses, and should therefore be considered for publication.

The description of methods used in Hierarchical Agglomerative Clustering (HAC) must be improved and additional tests must be added to accurately describe the level of confidence in these analyses. Specifically, the following points could be considered toward improvement:

A limitation of using HAC is that it always generates a cluster/dendrogram, regardless of the confidence in relationship of nodes/leaves. As a result, the dendrogram alone does not provide the complete result and must be supplemented with some kind of confidence metric or a p-value calculation. A useful tool for such analyses is the pvclust package:

Ryota Suzuki, Hidetoshi Shimodaira, Pvclust: an R package for assessing the uncertainty in hierarchical clustering, Bioinformatics, Volume 22, Issue 12, 15 June 2006, Pages 1540–1542, https://doi.org/10.1093/bioinformatics/btl117

More information about the clustering algorithm used (Ward, Median, etc.) and distance metric should be included in the methods section. The current description in methods is a very generalized outline and does not provide specific information pertinent to this analysis.

The manuscript could be considered for publication after the above points are addressed.

Author Response

Thank you for your considerations that certainly help us to improve our work. This study was designed to describe food-specific IgG concentrations and possible food-specific IgG clusters in a large cohort of subjects. We understand the limits of our hypothesis, and we hope that our results can be the basis for further analyses.

Thank you also for the suggestion to improve dendrogram. We agree with you about the methodological limit of an unsupervised hierarchical clustering algorithm. We hope to interpret your requests in the right way. So, we have added in the “methods” paragraph and in the caption of the figure a better description of the bar, to improve their comprehension. We have also added in the text and caption the p-value for every node, and for every node, the p-value is less than 0.001.

With these specifications, we have tried to improve the uncertainty in hierarchical clustering described in Suzuki’s paper you have proposed.  

As quoted in the text, we have used an adaptation of the statistical methods used by Eisen et al. We have not used the tools you have kindly suggested, but we hope that these new clarifications can be enough. For further studies in this field, we will certainly improve the dendrogram in accordance with the method you have proposed.

Linguistic note: before the submission to Nutrients and to you, we have proposed our paper to MDPI (the linguistic service of Nutrients) for revision and for plagiarism, accepting all their notes and comments and fixing them in accordance with their suggestions. 

We would like to thank you for your time and the attention given to us.  We remain at your disposal for any other suggestions.

Round  2

Reviewer 1 Report

The manuscript can be accepted in the revised form

Reviewer 2 Report

The authors have improved the manuscript based on the last review. While this is not a high-impact article, it would provide a good basis for future studies and therefore deserves to be published.